# Age-Related Changes in Serum *N*-Glycome in Men and Women—Clusters Associated with Comorbidity

**DOI:** 10.3390/biom14010017

**Published:** 2023-12-22

**Authors:** Óscar Lado-Baleato, Jorge Torre, Róisín O’Flaherty, Manuela Alonso-Sampedro, Iago Carballo, Carmen Fernández-Merino, Carmen Vidal, Francisco Gude, Radka Saldova, Arturo González-Quintela

**Affiliations:** 1Research Methodology Group, Health Research Institute of Santiago de Compostela (IDIS), Galician Health Service, University of Santiago de Compostela, 15706 Santiago de Compostela, Spain; oscar.lado.baleato@sergas.es (Ó.L.-B.); jorge.torre.eiriz@sergas.es (J.T.); manuela.alonso.sampedro@sergas.es (M.A.-S.); iago.carballo.fernandez@sergas.es (I.C.); carmen.fernandez.merino@sergas.es (C.F.-M.); carmen.vidal.pan@sergas.es (C.V.); francisco.gude.sampedro@sergas.es (F.G.); 2ISCIII Support Platforms for Clinical Research, Health Research Institute of Santiago de Compostela (IDIS), Galician Health Service, University of Santiago de Compostel, 15706 Santiago de Compostela, Spain; 3GlycoScience Group, National Institute for Bioprocessing Research and Training, Fosters Avenue, A94 X099 Dublin, Irelandradka.fahey@nibrt.ie (R.S.); 4Department of Chemistry, Maynooth University, W23 F2K8 Maynooth, Ireland; 5Primary Care, Santiago de Compostela Area, 15706 Santiago de Compostela, Spain; 6UCD School of Medicine, College of Health and Agricultural Science, University College Dublin, D04 V1W8 Dublin, Ireland

**Keywords:** *N-*glycome, aging, glycomics, public health, biomarker, diagnosis, serum

## Abstract

(1) Aim: To describe, in a general adult population, the serum *N-*glycome in relation to age in men and women, and investigate the association of *N-*glycome patterns with age-related comorbidity; (2) Methods: The serum *N-*glycome was studied by hydrophilic interaction chromatography with ultra-performance liquid chromatography in 1516 randomly selected adults (55.3% women; age range 18–91 years). Covariates included lifestyle factors, metabolic disorders, inflammatory markers, and an index of comorbidity. Principal component analysis was used to define clusters of individuals based on the 46 glycan peaks obtained in chromatograms; (3) Results: The serum *N-*glycome changed with ageing, with significant differences between men and women, both in individual *N-*glycan peaks and in groups defined by common features (branching, galactosylation, sialylation, fucosylation, and oligomannose). Through K-means clustering algorithm, the individuals were grouped into a cluster characterized by abundance of simpler *N*-glycans and a cluster characterized by abundance of higher-order *N-*glycans. The individuals of the first cluster were older, showed higher concentrations of glucose and glycation markers, higher levels of some inflammatory markers, lower glomerular filtration rate, and greater comorbidity index; (4) Conclusions: The serum *N-*glycome changes with ageing with sex dimorphism. The *N-*glycome could be, in line with the inflammaging hypothesis, a marker of unhealthy aging.

## 1. Introduction

Glycosylation is the most important post-translational modification of proteins and is key to their solubility, stability, and function [1]. Carbohydrates are rapidly becoming biomarker candidates because of their sensitivity to pathological changes in a wide array of diseases [2,3,4,5,6], particularly cancer [7,8,9,10,11]. It is nowadays possible to generate high-quality quantitative glycomic data in a high-throughput fashion [6,7,12].

The analysis of serum or plasma *N-*glycome consists of the determination of all glycans that are *N-*linked to asparagine in serum proteins. Some studies have addressed the changes in serum/plasma *N-*glycome with age, as recently reviewed [13]. The results have been partly heterogeneous due to the different techniques used and the different study populations. Some initial studies used DNA sequencing equipment-fluorophore assisted carbohydrate electrophoresis (DSA-FACE), which detects a limited number of *N-*glycan peaks (GP) and does not allow studying their sialylation [14,15,16,17,18,19]. Studies that use HPLC (high-performance liquid chromatography) variants detect a higher number of *N-*glycans and their sialylation [20,21,22,23], but, at the same time, the complexity of the results makes the analysis of the multiple data obtained and its interpretation difficult. The compositional nature of the data obtained must be considered in the statistical analyses, given that they are usually obtained as the area under the curve of a chromatogram in which the GPs sum up to 100%, i.e., relative quantitation; in that situation, the usual statistical tools may be used incorrectly. Another limitation of some previous investigations in order to delineate the effect of age on the *N*-glycome has been the samples studied, which have been highly selected in some cases, such as children and adolescents [22], people with Down’s syndrome [18], individuals with high longevity [24], or relatives of subjects with high longevity, in relation to control subjects [19,23]. The selection of the sample has been opportunistic in most studies (non-randomly based on the general population), and was based on volunteers [14,15,16,17,25,26]. To the best of our knowledge, the only population-based study was carried out on two islands of the Croatian coast as part of a program whose initial intention was to describe the variability and genetic epidemiology of closed island communities [20,21]. With these limitations, some common aspects of the findings of previous studies include a decrease over the years in biantennary, digalactosylated, and core-fucosylated *N-*glycans, along with an increase in the abundance of similar but agalactosylated *N-*glycans [14,15,16,17,18,19,21]. It is known that agalactosylated immunoglobulin-G (IgG) may participate in inflammatory disorders [27]. Moreover, the more precise study of *N*-glycosylation of immunoglobulin-G (IgG), the most abundant serum glycoprotein, may help to define a glycan clock to predict biological age [13,28,29,30,31,32]. Taken together, these findings are compatible with the notion of inflammaging (unhealthy inflammatory aging), so that changes in glycosylation could be both a marker of biological age and a factor in the causal chain of pathological aging [13,33,34,35,36].

Studies in randomly selected general populations could help the interpretation of abnormal laboratory results in the disease and the investigation of the potential influence of common determinants, such as demographic variables, lifestyle factors, common metabolic disorders, and comorbidities in general. The present study of the serum *N-*glycome using ultra performance liquid affinity chromatography (UPLC) includes a random sample of the general adult population with a wide age range to study the effect of age and the possible differences between men and women. As an approach to reduce dimensionality and thus facilitate the interpretation of serum *N-*glycome results, a principal component analysis (PCA) of *N-*glycan peaks obtained in 1516 individuals is presented.

## 2. Methods

### 2.1. Study Design and Setting

This cross-sectional study was developed in the municipality of A-Estrada (Spain, location 42°41′21″ N, 8°29′14″ W). An outline of the study (AEGIS, A-Estrada Glycation and Inflammation Study) is available at www.clinicaltrials.gov, code NCT01796184 and detailed elsewhere [37,38]. The municipality had an adult population (age >18 years) of 18,474 when the study started in 2012. An age-stratified random sample of the population aged 18 years and older was drawn from Spain’s National Health System Registry. From 2230 individuals that could be assessed for eligibility and displayed no exclusion criteria, 1516 agreed to participate (overall participation rate, 68%). From November 2012 to March 2015, all subjects were successively contacted and asked to attend the Primary Care Centre for evaluation, which included an interviewer-administered structured questionnaire and fasting venous blood sampling. Median age of participants was 52 years (range, 18–91 years) and 838 (55.3%) were women.

### 2.2. Ethical Issues

The general survey and the specific glycomic studies were approved by the Galician Regional Ethics Committee (codes 2010-315 and 2016-464, respectively) on 18 October 2016. Written informed consent was obtained from all participants.

### 2.3. Assessment of Smoking

Consumers of at least one cigarette per day were deemed to be smokers. Individuals who had quit smoking during the preceding year were still considered smokers.

### 2.4. Assessment of Alcohol Drinking

The habitual alcohol consumption was evaluated in standard drinking units [28], by summing the number of glasses of wine (~10 g), bottles of beer (~10 g), and units of spirits (~20 g) regularly consumed per week [39].

### 2.5. Usual Physical Activity History

All participants completed the Physical Activity Questionnaire (short version) [40], which has been validated in Spain [41]. The questionnaire allows the quantification of metabolic equivalents of task and the stratification of habitual physical activity into low, moderate, and high.

### 2.6. Definition of Metabolic Disorders

Obesity. The body mass index (BMI) was calculated with weight (in kg) divided by the square of height (in meters). Accordingly, participants were classified as normal weight (BMI < 25 kg/m^2^), overweight (BMI 25–30 kg/m^2^), or obese (BMI > 30 kg/m^2^).

Metabolic syndrome. Participants were considered to have metabolic syndrome when they met at least 3 criteria of the Adult Treatment Panel III (ATP-III) [42]: (1) abdominal obesity (waist circumference >102 cm for men and >88 cm for women); (2) hypertriglyceridemia (serum triglycerides ≥150 mg/dL); (3) low HDL-cholesterol levels (<40 mg/dL for men and <50 mg/dL for women); (4) elevated blood pressure (≥130/≥85 mmHg or current antihypertensive treatment); and (5) hyperglycemia (≥110 mg/dL or current antidiabetic treatment).

Diabetes mellitus. The diagnosis of diabetes was defined according to the American Diabetes Association criteria when the serum glycated haemoglobin (HbA1c) was ≥6.5% and/or fasting plasma glucose was ≥126 mg/dL or when the subject was receiving treatment with any antidiabetic [43].

### 2.7. Definition of Comorbidities and Their Quantification

A modification of the Charlson index [44] was used. Accordingly, diabetes mellitus, cerebrovascular disease, ischemic heart disease, peripheral arterial disease, heart failure, chronic renal failure, liver disease, cancer, chronic obstructive pulmonary disease, rheumatologic disease, and inflammatory bowel disease were considered comorbidities. The presence of each of them was awarded 1 point for summation purposes for quantification of comorbidity.

### 2.8. Routine Analytical Determinations

Serum concentrations of cholesterol, triglycerides, glucose, gammaglutamyl transferase (GGT), aspartate aminotransferase (AST), and creatinine were determined in fresh serum samples after morning fasting overnight on an automated ADVIA 2400 analyser (Siemens Healthcare Diagnostics, Barcelona, Spain) by the standard method recommended by the International Federation of Clinical Chemistry. To calculate the glomerular filtration rate, the MDRD-4 formula was used. The T3 hormone was determined by an immunochemometric assay on a Siemens Advia Centaur device (Siemens Health Care Diagnostics, Barcelona, Spain).

### 2.9. Determination of Inflammation Markers

C-reactive protein (CRP). Wide-range CRP was measured in fresh serum samples by a commercial latex-enhanced immunoturbidimetry method on an Advia 2400 Clinical Chemistry System (Siemens). Reference values with the method were 0–0.5 mg/dL. The measurement was available for 1499 individuals [45].

Erythrocyte sedimentation rate (ESR). ESR was measured in blood samples in K3EDTA tubes (Becton Dickinson, Franklin Lakes, NJ, USA) with a TEST-1 automatic apparatus (Alifax, Padova, Italy). The TEST-1 apparatus was validated against the Westergren reference method according to the criteria of the International Council for Standardization in Hematology. Reference values are 0–20 mm/h for men and 0–30 mm/h for women. ESR determination was available for 1472 participants [37].

Serum cytokines. Serum concentrations of interleukin (IL)-6, IL-8, TNF-alpha, and soluble IL-2 receptor (sIL-2R) were determined in fresh serum samples using a chemiluminescent immunoassay (IMMULITE 2000 System, Siemens Health Care Diagnostics, Barcelona, Spain). Results were available for 1499 individuals [46].

### 2.10. Determination of Glycation Markers (Markers of Glycaemic Control)

Glycated haemoglobin (HbA1c) was determined by liquid chromatography on an Adams A1C HA-8160 analyser (Arkray, Kyoto, Japan); HbA1c values were converted to units aligned with the Diabetes Control and Complications Trial, according to the US National Glycohemoglobin Standardization Program [47].

Fructosamine levels were determined by the diazyme glycated serum protein enzymatic method (Diazyme, Kent, UK) on an Advia 2400 analyser (Siemens). This assay uses protein K to digest serum proteins into small fragments and fructosaminase to catalyse the specific oxidation of the ketoamine linkage of glycated fragments. The release of hydrogen peroxide, measured colorimetrically at 550 nm, is proportional to the concentration of glycated serum proteins (fructosamine). To calculate the percentage of glycated albumin, the values of fructosamine (glycated serum proteins) were determined using the aforementioned method and those of albumin using the bromocresol green method. The glycated albumin percentage was calculated following the equation recommended by the manufacturer [48].

### 2.11. Serum N-Glycan Analyses

The serum *N-*glycome was analysed after thawing samples stored at −80 °C for further use. The complete procedure, developed by the authors as a high-throughput automated method [6], has been reported elsewhere [38], and is included as Appendix A (Appendix A). Briefly, *N-*glycans were released from the protein backbone enzymatically via Peptide:N-glycosidase F (PNGase F). Glycans were then immobilised on solid supported hydrazide beads and released from the solid supports and labelled with the fluorophore 2-aminobenzamide (2-AB) [49]. Hydrophilic interaction chromatography (HILIC) ultra-performance liquid chromatography (UPLC) provided chromatograms with 46 peaks [10]. The amount of glycans in each peak was expressed as % of total integrated area. Mass spectrometry-assisted glycan characterisation was performed for two representative samples; the major glycans were identified and assigned based on their glucose unit values cross-referenced in Glycobase, now migrated to GlycoStore, and based on previous assignments in Saldova et al. [10]. Glycan structures were annotated using the SNFG nomenclature and the DrawGlycan-SNFG software [50,51] with the assist of GlycoStore.org [52]. A summary of glycan peaks (GPs) and the corresponding *N-*glycan structures can be found in Appendix A. Furthermore, groups of GPs were defined from their common features, as follows [10]:

Galactosylation:

G0 (agalactosylated, GP1−2 + GP4−5 + (GP6/2) + (GP12/2));

G1 (monogalactosylated, GP3 + GP7−10 + (GP12/2) + GP16−18 + (GP21/2));

G2 (digalactosylated, GP13−15 + GP19 + GP20 + (GP21/2) + GP22−28);

G3 (trigalactosylated, GP29 + GP31−37); G4 (tetragalactosylated, GP30 + GP38−46).

Sialylation:

S0 (neutral (asialylated), GP1−15);

S1 (monosialylated, GP16−23 + GP30);

S2 (disialylated, GP24−29 + GP31);

S3 (trisialylated, GP32−40);

S4 (tetrasialylated, GP41−46).

Branching:

A1 (monoantennary, GP1−3 +(GP12/2) + (GP21/2));

A2 (biantennary, GP4−5 + (GP6/2) + GP7−10 + (GP12/2) + GP13−20 + (GP21/2) + GP22−28);

A3 (triantennary, GP29 + GP31−37);

A4 (tetraantennary, GP30 + GP38−46).

Oligomannose: (GP6/2) + GP11.

Fucosylation:

Core-fucose (CF) (GP2 + GP5 + (GP6/2) + GP8−10 + GP14−15 + GP17−18 + GP22−23 + GP27−28 + GP36 + (GP44/2)).

Outer-arm fucose (OF) (GP37 + GP40 + (GP41/3) + GP45 + (GP46/3)).

### 2.12. Statistical Analyses

Statistical tools appropriate to the nature of the inherent dependence of the data (Compositional Data Analysis techniques (CODA)) were used. CODA is based on the transformation of these constrained values into a set of isometric log-ratios (ilr). This ilr-transformed data can take any value in the real line; then, standard statistical techniques can be applied to them. Correlations between GP abundances were assessed by means of Spearman correlation coefficients obtained on the ilr-transformed values [53]. Differences between men and women were tested by applying a two-sample test for high-dimensional compositional data [54]. Then, in order to complete this multivariate test result, several non-parametric U-Mann–Whitney tests were applied on the ilr-transformed values in order to explore individual peak differences. Additionally, an ilr-transformed Generalized Additive Model (GAM) was estimated in order to assess age by gender effect on the *N-*glycans mean values. Regarding aggrupation of *N-*glycans by their common features, gactosylation, sialyation, and branching were treated as three sets of compositional data, and therefore CODA techniques were similarly applied to them. Oligomannose and core-fucosylation were treated as random variables with restricted domain (0,1), thus comparisons were also performed by means of non-parametric tests (Mann–Whitney test and Kruskal–Wallis test), and a beta response GAM. Robust principal component analyses [55,56] were performed in order to explore GP interdependence and to express their information in a more tractable way. For the definition of clusters and their analysis, an algorithm based on K-means was applied to the GP values transformed into ilr. The optimal number of clusters was defined graphically, based on the within sum of squares (elbow method) [57]. The groups thus defined were compared in their demographic and clinical characteristics using non-parametric tests (Mann–Whitney or Kruskal–Wallis test for continuous variables and Chi-square test for categorical variables). All P-values were corrected by the Benjamini method [58] in order to reduce the risk of false discovery rate. The probability of belonging to the clusters defined from serum *N*-glycome was studied using a multivariate logistic regression where age, sex, glycation markers, inflammation markers, and comorbidities were considered as predictor variables. Statistical analyses were performed in R (R core team, 2021), using the packages *compositional* [59], *compositions* [60], *robCompositions* [61], *mvoutlier* [62], and *mgcv* [63]. Graphical representations were made with the *ggplot2* package [64].

## 3. Results

### 3.1. N-Glycan Peaks (GPs) in Relation to Age and Sex

The multivariate compositional equivalence test showed that, overall, men and women do not have the same *N-*glycan profile (*p* < 0.001). The comparison of the 46 peaks of individual GPs between men and women, once the data were transformed into ilr coordinates, showed significant differences between men and women (Appendix A). Most of the differences, in one direction or the other, were observed in the higher order *N-*glycans (GP27 onwards, Appendix A).

The evolution of the abundance of the 46 GPs with age in men and women is represented in Figure 1. Appendix A further shows the age intervals in which there are statistically significant differences between men and women. The evolution and differences between sexes over age in the 46 GPs can be summarized into six patterns or groups, as follows (the *N*-glycan composition of each of the 46 GPs is represented in Appendix A):Group 1: GPs that are less abundant in women at younger ages, but increase in abundance over the years at a faster rate in this sex, so that at advanced ages their abundance becomes either equal or even higher than that of men. These are GP1, GP2, GP3, GP5, GP32, GP40, GP41, GP44, GP45, and GP46.Group 2: GPs that are more abundant in women at younger ages, but their abundance becomes equal or even higher in men in advances ages because they increase faster in men over the years (GP7, GP10, GP15, GP16, and GP23), or they decrease faster in women over the years (GP14, GP22, and GP24).Group 3: GPs that are similarly abundant in both sexes at young ages, but their abundance increases more prominently in women than in men over the years (GP39 and GP43).Group 4: GPs that are more abundant in one sex, regardless of age. They are GP29, GP30, GP31, GP33, GP34, GP36, and GP42 (more abundant in women), and GP35 and GP38 (more abundant in men).Group 5: GPs whose abundance increases or decreases over the years of age, but without clear differences between men and women. They are GP6, GP11, GP26, GP28, and GP37 (they increase over the years) and GP8, GP9, GP13, and GP18 (they decrease over the years).Group 6: GPs whose profile does not fit into any of the previous groups (GP4, GP12, GP17, GP19, GP20, GP21, GP25, and GP27).

Of note, examples of FAG2 (core-fucosylated, biantennary, and digalactosylated) *N-*glycans (GP14) decreased with age in both sexes, whereas examples of FA2 (core-fucosylated, biantennary, and agalactosylated), *N-*glycans (GP5) increased with age in both sexes (Figure 1). At young ages, GP14 abundance was slightly higher in women and GP5 abundance was slightly higher in men, but their abundance was similar in men and women at older ages (Figure 1 and Appendix A).

### 3.2. N-Glycan Groups (as Defined by Common Features) in Relation to Age and Sex

The comparison of the GP groups as defined by common features (galactosylation, sialylation, branching, fucosylation, and oligomannose) between men and women is presented in Appendix A. Regarding galactosylation, there were differences between men and women in the abundance of trigalactosylated (G3, higher in women) and tetragalactosylated (G4, higher in men) *N*-glycans. Similarly, females showed a higher abundance of triantennary (A3) *N-*glycans, while males showed a higher abundance of bi- and tetraantennary (A2 and A4) *N*-glycans. Regarding sialylation, the abundance of disialylated (S2) *N*-glycans was higher in men than in women, with no significant differences observed for the remaining sialylation groups. No significant differences were observed between men and women in the abundance of the different forms of fucosylation (neither core- nor outer-arm fucosylation) or oligomannose *N*-glycans (Appendix A).

The evolution with age of the abundance of the different *N*-glycan groups in men and women is represented in Figure 2 and Appendix A.

Taken together, the changes can be summarized into six patterns or groups, as follows:Group 1: An increase in abundance with age in men and women is observed for simple *N*-glycans such as agalactosylated (G0), monoantennary (A1), non-sialylated (S0, although not statistically significant in this case), and oligomannose (OM) *N*-glycans. The abundance of *N*-glycans with peripheral (outer-arm, OF) fucosylation also increase significantly with age, although it stabilizes after 50 years of age in both sexes.Group 2: A decrease in abundance with age in men and women is observed for digalactosylated (G2), biantennary (A2, although it stabilizes in men from middle age onwards), monosialylated (S1, although more significantly in women) and core-fucosylated (CF) *N*-glycans.Group 3: A trend to increase in abundance with age is observed for with tetrasialylated (S4), tetragalactosylated (G4), and tetraantennary (A4) *N*-glycans in women, while in men the abundance remains stable at a higher level, which tends to equal that of women at advanced ages.Group 4: An increase in abundance with age in both sexes is observed for trigalactosylated (G3) and triantennary (A3, with a non-significant trend in this case) *N*-glycans, although always at a higher level of abundance in women.Group 5: A divergent change in abundance with age between men and women starting at middle age (possibly coinciding with menopause in women) is observed for trisialylated (S3, which decrease in men with a continuous increase in women), and biantennary (A2, that stabilize at that age in men, with a continuous decline in women) *N*-glycans.Group 6: A difference in the abundance between men and women up to middle age is observed for agalactosylated (G0) *N*-glycans (more abundant in men), and monosialylated (S1) *N-*glycans (more abundant in women) (Figure 2).

### 3.3. Clusters Defined by Principal Component Analysis (PCA) and k-Means Clustering

After PCA, the variability of *N*-glycans (46 GPs) was adequately described by few components (dimensions) containing most of the variance of the original variables. Two dimensions explained, by themselves, more than 50% of that variance. The corresponding bar graph and biplot are represented Appendix A. The first dimension of the PCA contains, fundamentally, mainly information about simple *N*-glycans (mostly mono- or biantennary, non-sialylated, and agalactosylated or monogalactosylated *N*-glycans). These *N*-glycans are the first 11 GPs of the chromatographic spectrum and GP40 (Appendix A and Appendix A). A high proportion of those first 11 GPs is associated with a low value in this dimension. On the other hand, higher order *N*-glycans (GP 38, GP 40, and GP 44 to 46, the last in the chromatographic spectrum) were associated with a high value in this dimension (Appendix A). The interpretation of second dimension is more complex because it contains information from the last, higher order GPs of the chromatographic spectrum, but also information from GP2, GP3, GP13, and GP15 (Appendix A).

For the definition of clusters and their analysis, an algorithm based on K-means was applied to the GP values transformed into ilr. The optimal number of clusters (two) was established graphically and based on the sum of squares silhouette criteria (Figure 3A). The two groups of individuals thus defined are represented in Figure 3B, in relation to the two aforementioned dimensions of the PCA. The first group comprised 431 individuals, and the second 1085 individuals. The characteristics of the two clusters in relation to their *N*-glycan composition is represented in Table 1. Subjects in cluster 1 had a greater abundance of simple *N*-glycans. Specifically, they had a greater abundance of poorly branched *N*-glycans (A1 and A2), poorly galactosylated (G0 and G1), poorly sialylated (S0), and oligomannose (OM) *N*-glycans. Individuals of cluster 1 also had a higher abundance of core-fucosylated *N*-glycans (Table 1). On the contrary, subjects of cluster 2 had a greater abundance of high-order *N*-glycans. Specifically, they had a greater abundance of highly branched *N*-glycans (A3 and A4), highly galactosylated (G2, G3, and G4), highly sialylated (S2, S3, and S4) and *N*-glycans with peripheral fucosylation (outer-arm, OF) (Table 1). The composition of clusters 1 and 2 in terms of specific GPs is further represented graphically in Figure 3C and 3D, showing a predominance of simple *N*-glycans (GP1 to GP17) in cluster 1 and a predominance of higher order *N*-glycans (GP18 to GP46) in cluster 2.

Table 2 represents a comparison of demographic, lifestyle, metabolic, inflammatory, and comorbidity characteristics in the clusters defined from GPs of the serum *N*-glycome. No significant differences were observed in the sex distribution of cluster 1 and cluster 2, although the individuals in cluster 1 were significantly older those of cluster 2. Regarding lifestyle variables, no significant differences were observed in alcohol consumption and regular physical activity between both clusters, although the proportion of smokers was higher in the cluster 2. Individuals in cluster 1 presented data of metabolic disorders with greater frequency or quantity than those in cluster 2. The body mass index was similar in both clusters. The prevalence of diabetes mellitus and metabolic syndrome was slightly higher in cluster 1 (without statistically significant difference compared to cluster 2), but basal blood glucose was higher in the individuals in cluster 1, as were some glycation markers such as fructosamine and glycated albumin. No significant differences were observed in the inflammation markers between clusters 1 and 2, except for TNF-alpha, whose levels were significantly higher in the subjects in cluster 1. Some markers of liver damage, such as serum AST, were higher in cluster 1 individuals, and serum T3 hormone were lower in individuals from cluster 1. Similarly, individuals in cluster 1 had a lower glomerular filtration rate than subjects in cluster2. In line with the above, individuals in cluster 1 presented a higher rate of associated comorbidities than subjects in cluster 2 (Table 2).

The age- and sex-adjusted differences in laboratory determinations between individuals from cluster 1 (*n* = 431) and cluster 2 (*n* = 1085) are represented in Appendix A. After adjusting for age and sex, individuals from cluster 1 exhibited a 14.1 μmol/L higher level of fructosamine compared to those from cluster 2 (*p* < 0.001). Additionally, this group had a 0.58% greater level of glycated albumin (*p* < 0.001). Hepatic enzyme levels were also elevated after age and gender adjustment; AST levels were 0.96 IU/L higher (*p* = 0.034) and GGT levels were 5.34 IU/L higher (*p* = 0.003) for individuals of cluster 1. These individuals also had a 3.26 mL/min lower glomerular filtration rate (*p* = 0.008). No significant differences were observed in LDL levels or inflammation markers. While TNF-alpha levels were higher in men and increased linearly with age, no statistically significant differences between the clusters were found after adjusting for age and sex (Appendix A). The likelihood of experiencing two or more comorbidities is 53.1% higher in individuals from cluster 1, after age and sex adjustment (odds ratio 1.53; 95% CI 1.02–2.29). Differences in smoking status disappeared upon adjusting for age and gender. Finally, no differences were noted in physical activity or alcohol consumption between the clusters.

## 4. Discussion

The present study shows that the serum *N-*glycome changes significantly with age and that variations are different in men and women. In general, differences between men and women were more prominent for higher order *N-*glycans. Furthermore, certain *N-*glycome patterns are associated with older age, pathological changes, and comorbidity.

We observed that simple *N-*glycans such as monoantennary (A1), agalactosylated (G0) and oligomannose (OM) increase with aging in both sexes. The same trend is observed for the non-sialylated (S0) *N-*glycans. The decrease in galactosylation with aging had been previously reported and reviewed [13,35]. Initial studies using DSA-FACE reported an increase in core-fucosylated, biantennary, agalactosylated (FA2) *N-*glycans, together with a decrease in similarly core-fucosylated, biantennary, but digalactosylated, (FA2G2) *N-*glycans in samples of European [14,15,16,19] and Chinese [17] individuals. The findings were also observed in progeroid syndromes such as Werner syndrome [14], Cockayne syndrome [16] and Down’s syndrome [18]. In fact, the ratio between both *N-*glycans (glycan shift) has been called the GlycoAge test [16,18,19]. These results were confirmed in our study, in which we observed that FA2 structures (corresponding to GP5) increased with age, whereas FAG2 structures (corresponding to GP14) decreased with age. Moreover, two digalactosylated, biantennary *N-*glycans, both sialylated (feature not detected in DSA-FACE), one of them core-fucosylated (GP27) and one not (GP22), also decreased with age. The results of the Croatian group, of greater comparability with ours in terms of study design and technique used (liquid chromatography), also showed an increase in agalactosylated *N-*glycans and a decrease in digalactosylated *N-*glycans with age [21]. Some previous studies reported a greater decrease with age of the FAG2-type *N-*glycans in women compared to men [17,21,23], which was potentially attributed to the decrease in oestrogen in women after the menopause, give the well-known effect of oestrogens on glycosylation [29,65]. This was not exactly confirmed in our study, but we observed that, at younger ages, FAG2-type *N-*glycan (GP14) abundance was higher in women than in men, whereas that FA2-type *N-*glycan (GP5) abundance was higher in men, but their abundance decreases with ageing in both sexes and became similar in men and women at older ages.

The increase in abundance of agalactosylated (G0) *N-*glycans with age has also been observed in the specific analysis of IgG glycosylation and in immune-based diseases [13,31,32,35]. These data support the notion of an association of changes in *N-*glycosylation with the so-called inflammaging, that is, harmful aging conditioned by inflammatory activity [13,31,32,35]. Chronic low-grade inflammation during aging, without infection or other overt inflammatory stimuli (inflammaging), is associated with increased morbidity and mortality in the aging population [66]. In a hypothetical scheme, the glycosylation machinery of the antibody-producing cells would be altered with ageing, and this would result in the increased expression of agalactosylated and poorly sialylated IgG. This aberrantly glycosylated IgG would activate some effector branches of immunity (complement, presenting cells, phagocytes), which would result in the amplification of inflammatory signals. According to this model, agalactosylated *N-*glycans, especially in IgG, would be a key factor for an inflammatory amplification loop and not merely a marker of inflammation and age [35,36]. Indeed, alterations in *N-*glycosylation have been proposed as a tool for measuring biological age as opposed to chronological age. As already mentioned, changes in the serum/plasma *N-*glycome have been reported as the GlycoAge test [14,15,16], and more precise changes of IgG glycosylation have given rise to a commercial test that aim to quantify biological age (see https://glycanage.com/, accessed on 20 December 2023).

The rest of the alterations of isolated GPs and their common features in serum/plasma *N-*glycome have been less studied in the literature. To the best of our knowledge, the increase in abundance with age of oligomannose *N-*glycans (corresponding GP6 and GP11 in our study), as observed for both sexes, was not previously described. A recent review has highlighted the role of oligomannose *N-*glycans in some diseases with defective protein folding, including several age-related diseases [67].

We observed that higher-order *N-*glycans, such as tetra-antennary (A4) and tetragalactosylated (G4), were more abundant in men, but their abundance increased with age in women until they become similarly abundant at advanced ages. This finding was not confirmed in the Croatian study [21] (the most comparable to ours in terms of general population design, glycomic technique, and sample size), in which the abundance of these *N-*glycans A4 and G4 was slightly higher in women. In that study, A4 *N-*glycans tended to increase with age, especially in women, while G4 tended to decrease with age in both sexes [21]. Similarly, Ding et al. (2011) also reported a greater abundance of A4 in women [17]. It must be considered, however, that the abundance of this form of *N-*glycans is low. In our study, trigalactosylated (G3) and triantennary (A3) *N-*glycans were more abundant in women than in men, with little variation with age, except for a significant increase in G3 in women. These findings are similar to those observed by Knezević et al. (2010) [21]. In a small series of centenarian Japanese individuals, a high abundance of multi-antennary (and multi-sialylated) *N-*glycans was detected compared to elderly controls and younger subjects [24]. The abundance of higher order *N-*glycans has been also observed in inflammatory diseases [68]. In this aspect, referring specifically to the tetra-antennary (A4) and tetragalactosylated (G4) forms, men would have higher abundance than women until very advanced ages, in which both sexes would tend to equalize.

Fucosylation of proteins is an important element in health and disease [69]. In our study we have observed that core fucosylation decreases similar with age in both sexes. This finding is consistent with those of previous studies [14,21,22,23]. Peripheral (outer-arm) fucosylation, on the contrary, increases until middle ages of life and stabilizes thereafter, also in both sexes, with a slight predominance in women. The findings are slightly different from those reported in the aforementioned Croatian population [21], in which peripheral fucosylation was, overall, higher in men, with a tendency to increase with age in men at even to decrease in women. It should be considered that, in circumstances such as this where changes with age do not follow a linear course, simply estimating the age effect with a linear correlation [21] can provide misleading results.

Sialylation of proteins is important in many pathological processes [70]. The level of total plasma sialic acid (bound and free) increases with age [25]. However, the situation is not so clear with respect to sialylated *N-*glycans attached to plasma proteins, the complexity of which decreases with age in children [22] and increases in people of extreme longevity [24]. According to our data, sialylation is the most irregular glycosylation phenomenon in relation to age. This is not surprising [13], because serum sialylated *N-*glycans originate from a large number of diverse proteins [71], many of which can vary in their concentration in response to the physiological or pathological states, such as acute phase proteins and immunoglobulins [13]. As already mentioned, desialylated (S0) *N-*glycans tend to increase with age, in a non-significant manner and without difference between men and women, which is consistent with the aforementioned decrease in *N-*glycans without galactose, to which sialic acid residues are attached. In monosialylated *N-*glycans (S1), a tendency to decrease with age was observed in women, but it was not significant in men, similar that observed by Knezević et al. (2010) [21], without global differences between both sexes, as in the aforementioned study. The behaviour of disialylated *N-*glycans (S2) was also similar to that reported in previous studies [21], with a higher global abundance level in men than in women, and a tendency to decrease with age in men and increase in women. Regarding trisialylated (S3) and tetrasialylated (S4) *N-*glycans we observed differences between men and women starting at middle age, partly coinciding with menopause in women. The trisialylated (S3) *N-*glycan abundance decreased from that age in men, with a continuous increase in women. Tetrasialylated (S4) *N-*glycan abundance increased gradually with age in women until advanced ages, whereas it remained stable in men at all age ranges. Regarding S3 *N-*glycans, the Croatian group also did not observe significant overall differences between men and women and, similar to our data, a positive correlation with age was observed in women [21]. Regarding the S4, in the study by Knezević et al. (2010), no such positive correlation with age was observed in women [21]. This possible hormonal effect on sialylation was not, to our knowledge, described. Overall, in the study of the Croatian group, more variations with age were found in women than in men, which was also attributed to the possible variable hormonal effect throughout life [21]. In our experience, with the aforementioned reservations, there was a similar number of variations with age in men and women.

The preceding paragraphs illustrate the complexity of serum *N-*glycome results, which makes their interpretation difficult. Our study investigated the possible utility of PCA as a tool to reduce *N-*glycome dimensionality and simplify *N-*glycome interpretation. According to a K-means clustering algorithm of the 46 *N*-glycan peaks of the chromatogram, individuals can be grouped into two main clusters, the first of which is fundamentally characterized by a higher abundance of the simplest *N-*glycans (GP 1 to 11, which are agalactosylated or monogalactosylated, monoantennary or biantennary, and asialylated, including oligomannose). Some of these simple *N-*glycans are known to be associated with ageing and inflammation, as mentioned above. The second cluster is fundamentally defined by the higher order *N-*glycans and a lower abundance of simple *N-*glycans. Importantly, individuals in the cluster associated with the simplest *N-*glycans are older, have higher levels of blood glucose and glycation markers, higher levels of some inflammation markers, higher concentrations of AST, lower concentrations of T3 hormone, lower glomerular filtration rate, and a higher comorbidity rate. Although the individuals in the first cluster are older than those in the second cluster, the association with comorbidity and markers of common metabolic disorders markers was independent of age. In line with this, age is associated with higher values of some inflammation markers in this population, such as erythrocyte sedimentation rate (ESR), serum IL-6, and sIL-2r [37,45,46]. Taken together, the results suggest that serum *N-*glycome, analysed by this synthetic technique, could be, in line with the inflammaging hypothesis, a marker of unhealthy aging. To our knowledge, this global approach to *N-*glycome has not been used previously.

The study has limitations that should be acknowledged. The study of *N-*glycome is a general approach that does not allow us to know which specific proteins are those that present changes in glycosylation, as opposed to studies focused on abundant glycoproteins with immune involvement such as IgG. However, the study of the *N-*glycome allows a general view of the glycosylation status of all serum/plasma glycoproteins. The general population-based design may be considered a strength of the study. To our knowledge, this is the first study of serum *N-*glycome on a randomly selected general adult population, which increases its representativeness and reduces potential selection biases. Temporal ambiguity is inherent to the cross-sectional design when investigating cause–effect association. Future longitudinal studies will be necessary to better establish whether the reported changes in serum *N-*glycome can be a prognostic marker of unhealthy aging and mortality.

## 5. Conclusions

1. The serum *N*-glycome changes substantially with age; changes are, in part, different in men and women.

2. In both sexes, simple *N*-glycans such as agalactosylated (G0, which have been found related to inflammation), mono-antennary (A1), oligomannose (OM), and non-sialylated (S0) tend to increase with age.

3. In both sexes, core fucosylation decreases with age; peripheral fucosylation (outer-arm) increases until middle ages of life and stabilizes thereafter.

4. There is sexual dimorphism in the evolution with age of higher order *N-*glycans with high levels of branching, galactosylation, and sialylation.

5. Through K-means clustering of the 46 *N-*glycan peaks of the chromatogram, individuals can be grouped into two main clusters, the first of which is characterized by the simplest *N-*glycans and the second by the higher order ones. Individuals in the cluster associated with the simplest *N-*glycans are older, have higher levels of blood glucose and glycation markers, higher levels of some inflammation markers, lower glomerular filtration rate, and greater comorbidity.

6. Taken together, the results are consistent with the idea that serum *N*-glycome could be, in line with the inflammaging hypothesis, a marker of unhealthy inflammatory aging.

## Figures and Tables

**Figure 1 biomolecules-14-00017-f001:**
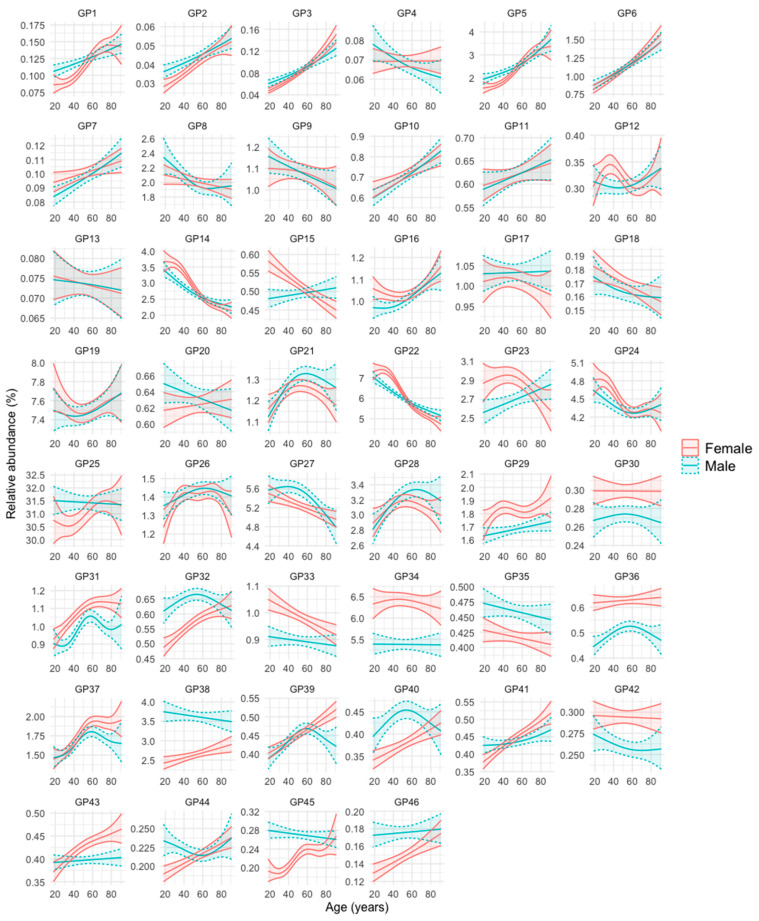
Glycan peak (GP) abundance in relation to age and sex. For each GP, the mean relative abundance (in percentage) with its 95% confidence interval in men and women was obtained after transforming the data into ilr (isometric log-ratios). The curves were obtained using generalized additive regression models with a predictor factor per curve, estimating a smooth effect of age (using spline functions) for men and women. The adjustments were obtained on the ilr scale and the results were expressed on the real scale after applying its inverse function.

**Figure 2 biomolecules-14-00017-f002:**
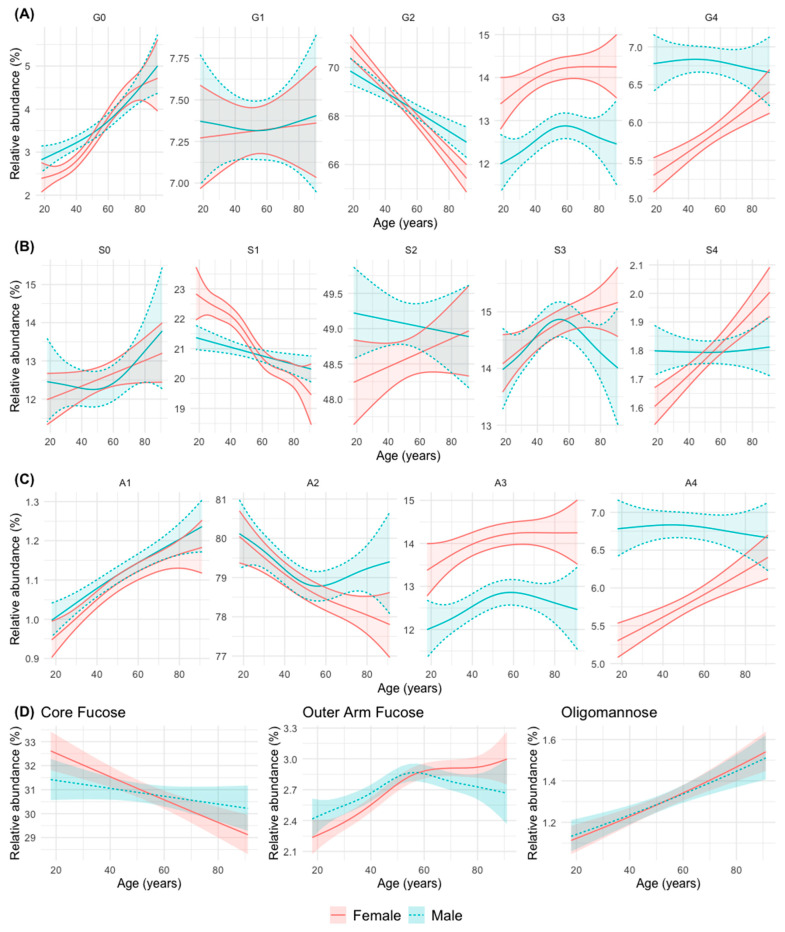
Abundance of groups of *N-*glycans according to their common features: (**A**) galactosylation (G0 a-, G1 mono-, G2 di-, G3 tri-, and G4 tetragalactosylated); (**B**) sialylation (S0 a-, S1 mono, S2 di-, S3 tri-, and S4 tretrasialylated) (**C**) branching (A1 mono-, A2 bi-, A3 tri-, andA4 tetraantennary); (**D**) fucosylation (core- and outer-arm fucosylation) and oligommannose in relation to age in the men and women. The mean relative abundance (in percentage) is represented, with its 95% confidence interval. The curves were obtained using generalized additive regression models with a predictor factor per curve, estimating a smooth effect of age (using spline functions) for men and women. For galactosylation, branching, and sialylation, the adjustments were obtained on the isometric log-ratio (ilr) scale and the results were expressed on the real scale after applying its inverse function. For fucosylation and oligomannose, the regression response was modelled as a beta distribution variable (bounded between 0 and 1).

**Figure 3 biomolecules-14-00017-f003:**
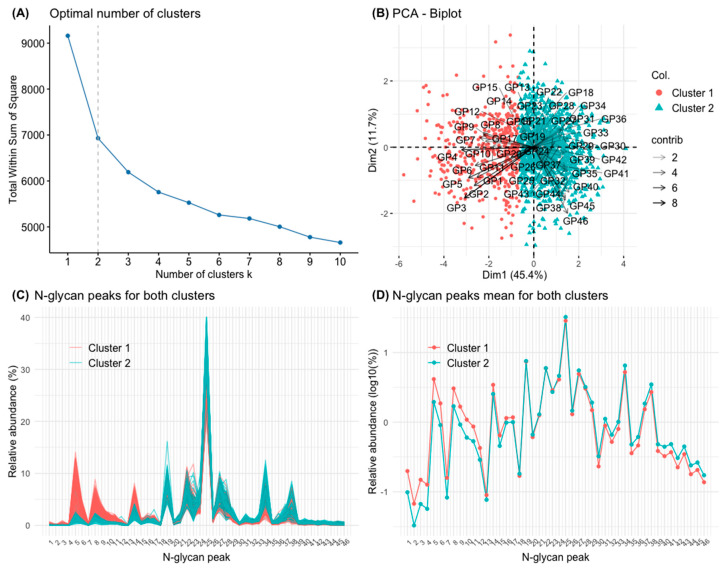
Principal component analysis (PCA) of the 46 glycan peaks (GPs). (**A**) Elbow graph to determine the optimal number of groups (clusters). Elbows are observed when the number of groups is 2 and 4. For the present work, two groups (clusters) have been defined. (**B**) PCA biplot with the clusters defined by K-means of the set of GPs. (**C**) The 1516 chromatograms with their 46 GPs are represented individually, separated in different colours depending on whether they belong to the cluster 1 or 2, respectively. (**D**) The means of the 46 GPs in the study population are represented (on a logarithmic scale), separated in different colours depending on whether they belong to cluster 1 or 2, respectively.

**Table 1 biomolecules-14-00017-t001:** Serum *N*-glycan trait abundance in the two clusters defined by principal component analysis of serum glycan peaks.

*N*-Glycan Trait (%)	Total Sample (*n* = 1516)	Cluster 1 (*n* = 431)	Cluster 2 (*n* = 1085)	*p*-Value
G0	3.19 (2.40, 4.53)	5.75 (4.47, 7.92)	2.76 (2.20, 3.41)	<0.001
G1	6.95 (6.05, 8.26)	9.59 (8.44, 11.60)	6.41 (5.72, 7.14)	<0.001
G2	67.98 (65.43, 69.98)	64.98 (61.46, 68.21)	68.57 (66.82, 70.45)	<0.001
G3	13.48 (11.54, 15.41)	11.28 (9.83, 12.95)	14.30 (12.52, 15.92)	<0.001
G4	6.22 (5.09, 7.58)	5.22 (4.29, 6.23)	6.66 (5.62, 7.89)	<0.001
A1	1.08 (0.95, 1.22)	1.30 (1.16, 1.49)	1.01 (0.91, 1.12)	<0.001
A2	77.76 (75.70, 79.82)	80.05 (78.34, 81.48)	76.92 (75.07, 78.63)	<0.001
A3	13.48 (11.54, 15.41)	11.28 (9.83, 12.95)	14.30 (12.52, 15.92)	<0.001
A4	6.22 (5.09, 7.58)	5.22 (4.29, 6.23)	6.66 (5.62, 7.89)	<0.001
S0	11.83 (9.82, 14.69)	17.92 (15.23, 23.93)	10.66 (9.26, 12.08)	<0.001
S1	21.15 (19.48, 22.85)	21.37 (19.59, 23.27)	21.08 (19.45, 22.63)	0.085
S2	49.50 (47.16, 51.29)	45.02 (42.05, 47.60)	50.39 (49.00, 52.03)	<0.001
S3	14.87 (13.06, 16.59)	12.57 (11.17, 13.96)	15.68 (14.35, 17.15)	<0.001
S4	1.82 (1.50, 2.13)	1.48 (1.26, 1.76)	1.93 (1.68, 2.23)	<0.001
CF	30.44 (27.23, 33.93)	35.68 (32.65, 39.53)	28.76 (26.09, 31.45)	<0.001
OF	2.67 (2.28, 3.12)	2.30 (1.96, 2.61)	2.83 (2.45, 3.25)	<0.001
OM	1.10 (0.91, 1.40)	1.76 (1.40, 2.65)	0.99 (0.86, 1.15)	<0.001

Data are medians and interquartile ranges (in parentheses). The raw data are represented (in percentage of abundance), although for statistical comparisons they were previously transformed into isometric log-ratios (ilr). The fucosylation and oligomannose traits were treated as random variables with restricted domain (0,1). *p*-values were obtained with the Mann–Whitney test with Benjamini correction. G, galactosylation (G0 a-, G1 mono, G2 di-, G3 tri-, and G4 tetra-galactosylated). A, antennae (branching: A1 mono-, A2 di-, A3 tri-, A4 tetra-antennary). S, sialylation (S0 a-, S1 mono-, S2 di-, S3 tri-, and S4 tetra-sialylated). F, fucosylation (CF, core-fucosylation; OF, outer arm fucosylation). OM, oligomannose.

**Table 2 biomolecules-14-00017-t002:** Comparison of demographic data, lifestyle factors, metabolic variables, inflammatory markers and comorbidity index in the two clusters defined by principal component analysis of serum glycan peaks.

	Cluster 1 (*n* = 431)	Cluster 2 (*n* = 1085)	*p*-Value
Age, years	56 (40, 70)	51 (38.00, 65)	0.002
Women, n (%)	232 (53.8)	606 (55.8)	0.608
Men, n (%)	199 (46.2)	479 (44.2)	
Smoking status			
Never smokers, n (%)	237 (55.0)	588 (54.2)	0.031
Ex-smokers, n (%)	128 (29.7)	267 (24.6)	
Smokers, n (%)	66 (15.3)	230 (21.2)	
Physical activity			
Low, n (%)	165 (38.3)	431 (39.7)	0.891
Medium, n (%)	161 (37.3)	391 (36.0)	
High, n (%)	105 (24.4)	263 (24.2)	
Alcohol consumption (g/day)			
0–9, n (%)	159 (36.9)	387 (35.7)	0.468
10–139, n (%)	162 (37.6)	436 (40.2)	
140–279, n (%)	78 (18.1)	163 (15.0)	
≥280, n (%)	32 (7.4)	99 (9.1)	
Body mass index, kg/m^2^	27.7 (24.5, 31.5)	27.7 (24.6, 31.3)	0.726
Diabetes mellitus, n (%)	61 (14.2)	126 (11.6)	0.358
Metabolic syndrome, n (%)	95 (22.0)	219 (20.2)	0.608
Serum glucose, mg/dL	91 (83, 102)	88 (81, 98)	0.006
Blood glycated hemoglobin (HbA1c), %	5.4 (5.2, 5.8)	5.4 (5.2, 5.7)	0.752
Serum glycated albumin, %	14.1 (13.0, 15.5)	13.6 (12.4, 14.8)	<0.001
Serum fructosamine, μmol/L	262 (237, 292)	251 (220, 279)	<0.001
Serum HDL-cholesterol, mg/dL	59 (49, 71)	57 (46, 68)	0.006
Serum LDL-cholesterol, mg/dL	114 (93, 137)	113 (94, 134)	0.905
Erythrocyte sedimentation rate (ESR), mm/h	9 (5, 17)	9 (5, 16)	0.326
Serum C-reactive protein, mg/dL	0.13 (0.04, 0.33)	0.14 (0.05, 0.41)	0.358
Serum TNF-alpha, pg/mL	7.6 (6.4, 9.6)	7.4 (6.0, 8.8)	0.001
Serum interleukin-8 (IL-8), pg/mL	7.0 (5.0, 10.0)	7.0 (5.0, 11.0)	0.608
Serum interleukin-6 (IL-6), pg/mL	2.1 (2.0, 3.2)	2.2 (2.0, 3.6)	0.358
Serum soluble interleukin-2 receptor, U/mL	414 (321, 527)	411 (313, 530)	0.608
Serum aspartate aminotransferase (AST), IU/L	23 (20, 28)	22 (19, 27)	0.002
Gamma-glutamyl transferase (GGT), IU/L	21 (14, 34)	19 (13, 33)	0.069
Serum triiodothyronine (T3), pg/mL	3.30 (3.05, 3.56)	3.39 (3.15, 3.67)	0.001
Glomerular filtration rate, mL/min/1.7 m^2^	96.8 (84.5, 111.8)	101.9 (88.3, 117.2)	0.001
Comorbidity index			
0 points	296 (68.7)	789 (72.7)	0.015
1 point	84 (19.5)	222 (20.5)	
≥2 points	51 (11.8)	74 (6.8)	

Data are medians and interquartile ranges (in parentheses) or absolute numbers and percentages (in parentheses). *p*-values were obtained with the Mann–Whitney test or the chi-squared test, respectively, with Benjamini correction. LDL, low-density lipoprotein; HDL, high-density lipoprotein; TNF, tumour necrosis factor.

## Data Availability

The data that support the findings of this study are available on reasonable request from the corresponding author. The data are not publicly available due to Spanish law restrictions.

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
