# Peer review of "Age-Related Changes in Serum N-Glycome in Men and Women—Clusters Associated with Comorbidity"

_biomolecules, 2023, doi:10.3390/biom14010017_

Round 1
Reviewer 1 Report
Comments and Suggestions for Authors
The manuscript by Lado-Baleato aimed at investigating age/gender differences in serum N-glycome in a large cohort of individuals with detailed health characteristics. This excellent cohort included >1500 individuals with detailed information regarding age, gender, smoking, alcohol consumption, physical activity, metabolic disorders, comorbidities (e.g. diabetes, CVD, cancer etc), blood analytic and inflammation markers. N-glycans were examined by standard HILIC-UPLC and normalized values grouped into characteristic summary glycan peaks, then correlation with health/age/gender investigated by compositional analysis techniques. Initial analysis revealed some differences in N-glycan peaks according to age and sex. This motivated further analysis of grouped N-glycans according to common their features, with subsequent PCA analysis and k-mean clustering. The discussion is comprehensive and puts the findings in context with previous studies, some corroborating what was previously known, together with other novel findings from this current analysis. Overall, this is a useful study, very well designed and comprehensive, and the analytical tools developed here puts the findings into a manageable perception of the factors that affect changes in serum N-glycans.
Major comments:
1. It would be useful to have a summary figure to highlight major findings
Minor comments:
1. The legend in some figures are too small or not well positioned
2. Supplementary methods need clarifications on the temperature used for PNGase-F treatment, and detailed description of 2AB labeling is also missing.
Author Response
Major comments:
- It would be useful to have a summary figure to highlight major findings.
R: A summary figure (Supplementary Figure 3) has been includedin the revised version, according to the reviewer suggestion.
Minor comments:
- The legend in some figures are too small or not well positioned.
R: The legend for Figure 1 has been made larger. The legend of Figure 2 has been repositioned to make it more visible.
- Supplementary methods need clarifications on the temperature used for PNGase-F treatment, and detailed description of 2AB labelling is also missing.
R: As described in the revised Supplementary methods, the assay was performed at room temperature. For the detailed description of 2AB labelling, a reference from our group has been added.
Reviewer 2 Report
Comments and Suggestions for Authors
In this manuscript, Lado-Baleato et. al analyze the N-glycan serum profile of 1516 adults in a randomized study using HILIC-UPLC. They describe the glycoprofiles in the context of age and sex and attempt to correlate them to various comorbidities through a K-means clustering algorithm. They observe statistically significant differences between men and women of different ages and identify a cluster with lower order N-glycans that could be linked to some comorbidities, generally in agreement with previous studies.
The primary strength of the paper is the sample set – randomly selected individuals in a general population – to which the glycan analysis is applied.
While the scope of the paper is limited and provides a statistical correlation study without further investigation into specific proteins or mechanisms, it may serve as a foundation for future studies.
It is unclear whether the six groupings of N-glycan peaks (GPs) into which the authors group them are biologically meaningful in any way. Regardless, the data could be more easily navigable if Figure 1 were organized according to the six groupings (e.g., place GP1, GP2, GP3, GP5, GP32, GP40, GP41, GP44, GP45 and GP46 together and label as “Group 1”) rather than displaying them in numerical order.
One major issue is that the data reported in Table 2 does not seem to be corrected for age and sex. This correction is mentioned in the last paragraph of the results regarding fructosamine, comorbidity score, and TNF-alpha but the numbers don’t seem to be the same as those in Table 2. Because the average age of individuals of cluster 1 is higher, it would be beneficial to also report the age-adjusted data for all the criteria since the majority of them can be primarily correlated to age instead of glycosylation profiles.
Comments on the Quality of English LanguageQuality of English is generally fine but some minor editing could improve it.
Author Response
Comments:
- It is unclear whether the six groupings of N-glycan peaks (GPs) into which the authors group them are biologically meaningful in any way. Regardless, the data could be more easily navigable if Figure 1 were organized according to the six groupings (e.g., place GP1, GP2, GP3, GP5, GP32, GP40, GP41, GP44, GP45 and GP46 together and label as “Group 1”) rather than displaying them in numerical order.
R: Groupings N-glycans make sense because they allow them to be combined based on their common characteristics, in order to better interpret the complexity that would be implied by analyzing the 46 peaks separately. This strategy has been commonly used by our group [Saldova et al., 2014, reference number 10] and other groups [Knezevic et al., 2010, reference number 21]. Unfortunately, it is not possible to present the 46 glycan peaks (GPs) ordered by these groupings, since the same GP can belong to several of them. For example, GP2 corresponds to an agalactosylated, monoantennary, asialylated, and core-fucosylated N-glycan.
- One major issue is that the data reported in Table 2 does not seem to be corrected for age and sex. This correction is mentioned in the last paragraph of the results regarding fructosamine, comorbidity score, and TNF-alpha but the numbers don’t seem to be the same as those in Table 2. Because the average age of individuals of cluster 1 is higher, it would be beneficial to also report the age-adjusted data for all the criteria since the majority of them can be primarily correlated to age instead of glycosylation profiles.
R: In the revised version, the paragraph explaining the differences between cluster 1 and 2 that are maintained when adjusting for age and sex has been expanded. In addition, a new supplementary table (number 5) has been included. This table displays the differences in the quantitative variables between clusters 1 and 2 of Table 2, after adjusting for age and sex.
Reviewer 3 Report
Comments and Suggestions for Authors
The current manuscript deals with a very complex problem i.e. evolution of glycans in relation with age. In an overall manner the results described herein can be assessed as preliminary but really deserved to be reported in a publication. Improvements will require tens of years and new means of analytical studies. In conclusion, a very interesting study !
Author Response
R: No comments to make. The authors thank the reviewer comments.